# Leishmaniasis Vaccines: Applications of RNA Technology and Targeted Clinical Trial Designs

**DOI:** 10.3390/pathogens11111259

**Published:** 2022-10-29

**Authors:** Malcolm S. Duthie, Bruna A. S. Machado, Roberto Badaró, Paul M. Kaye, Steven G. Reed

**Affiliations:** 1HDT Bio, 1616 Eastlake Ave E, Seattle, WA 98102, USA; 2SENAI Institute of Innovation (ISI) in Health Advanced Systems (CIMATEC ISI SAS), University Center SENAI/CIMATEC, Salvador 41650-010, Bahia, Brazil; 3York Biomedical Research Institute, Hull York Medical School, University of York, York YO10 5DD, UK

**Keywords:** leishmania, vaccine, RNA, clinical trials

## Abstract

*Leishmania* parasites cause a variety of discrete clinical diseases that present in regions where their specific sand fly vectors sustain transmission. Clinical and laboratory research indicate the potential of immunization to prevent leishmaniasis and a wide array of vaccine candidates have been proposed. Unfortunately, multiple factors have precluded advancement of more than a few *Leishmania* targeting vaccines to clinical trial. The recent maturation of RNA vaccines into licensed products in the context of COVID-19 indicates the likelihood of broader use of the technology. Herein, we discuss the potential benefits provided by RNA technology as an approach to address the bottlenecks encountered for *Leishmania* vaccines. Further, we outline a variety of strategies that could be used to more efficiently evaluate *Leishmania* vaccine efficacy, including controlled human infection models and initial use in a therapeutic setting, that could prioritize candidates before evaluation in larger, longer and more complicated field trials.

## 1. Introduction

*Leishmania*, obligate intracellular macrophage parasites, are an important genus of parasites that can affect humans and canines for which disease manifestation is dependent upon the infecting parasite species. Each *Leishmania* species demonstrates a geographic range that is naturally determined by the presence of their specific sand fly vector, and thus differing forms of leishmaniasis are dispersed across endemic regions. Given migration and that the parasites can be transferred in blood, however, cases are occasionally observed in non-endemic regions. Most infections remain asymptomatic (e.g., 90% of humans infected with *L. donovani* do not advance to symptoms) with a key element for parasite containment being an effective antigen-specific T cell response that can prevent advancement to, or resolution from, the diseased state [1,2]. Recovery from primary infection is typically associated with long term protection against reinfection, indicating the potential for generating lasting protection through the use of durable anti-*Leishmania* vaccines [1,2].

In addition to their own clinical significance, experimental *Leishmania* infection models have served as an important tool for general immunological understanding. Seminal work in the 1980s used these models to define the Th1/2 paradigm, identifying that experimental *L. major* infection becomes established then typically clears as antigen-specific Th1 cells develop in resistant C57BL/6 mice whereas infection continues unabated despite the Th2 cells that predominate in the susceptible BALB/c mice [3,4]. These classic models have also been used to define numerous subsets of CD4^+^ T cells and reveal genetic mechanisms involved in the development of both disease and adaptive immunity [5,6,7]. Beyond CD4^+^ T cells, there is evidence that CD8^+^ T cells also participate in protection in both experimental and physiological situations [8,9]. Thus, *Leishmania* infection provides a strong basis for the evaluation of T cell-inducing vaccines and determining their durability.

## 2. Potential Application of RNA Technology for Leishmania Vaccines

An increase in the clinical availability and use of antileishmanial drugs has been observed in recent years, with a positive impact observed in the reduced severity of disease especially in the most lethal visceral leishmaniasis manifestation[10]. Such treatment of leishmaniasis still presents with the classical challenges of any drug treatments, including the emergence of drug resistant parasites[11], however, and prevention through immunization appears both attainable and preferred. Unfortunately, despite the plethora of preclinical evaluations multiple factors have precluded advancement of more than a few *Leishmania* targeting vaccines to clinical trial. The sheer volume of potential vaccine platforms and antigen targets identified, the substantial economic impact of producing these as GMP-grade, and the lack of availability of safe and effective adjuvants with which to enhance or sustain responses, has led to reticence in advancing to trial candidates that appear to have ‘room for improvement’ [12,13,14,15]. Further, given that infection is reliant on sand fly vector transmission and sand fly populations are impacted by seasonal variation and micro-geography, pre-planning for enactment in regions with sufficiently high parasite transmission at time of trial is both difficult and unassured. The risk of over-estimating infection rates and inadvertently under-powering *Leishmania* vaccine trials that have disease prevention as an endpoint may be mitigated by a predictive controlled human infection model (CHIM).

RNA-based vaccines rapidly emerged in response to the COVID-19 pandemic largely because (a) the SARS-CoV-2 Spike protein was rationally selected as the target antigen given previous work on SARS and MERS [16], and (b) RNA vaccines could be made and released at GMP grade far more rapidly than subunit vaccines involving recombinant proteins. The inserted RNA sequence can be modified with relative ease and as the COVID-19 pandemic has continued with the evolution to SARS-CoV-2 variants-of-concern, vaccines have accordingly been quickly updated to allow evaluation of revised Spike antigen sequences [17,18,19]. Applying the same logic to *Leishmania*, where multiple antigens are known to afford at least some protection in animal models, it may be possible to evaluate leads, then quickly revise them, in response to emerging clinical data. Relative ease in the design and manufacture of nucleic acid-based vaccines also suggests the potential for inexpensive and somewhat generic production. One considerable logistical advantage of RNA-based vaccines over the majority of other platforms is that the RNA can be produced in a cell-free environment by in vitro transcription, removing the need for cultured cells in the manufacturing process and avoiding the quality and safety issues associated with their use. In this way, it is possible to perform simple downstream purification to provide both faster and more cost-effective manufacturing, and robust manufacturing processes have now been established for both mRNA and self-amplifying replicon RNA (repRNA) constructs. Thus, RNA vaccines possess an inherent nimbleness that recombinant proteins of defined subunit vaccines do not. Further, the composition of T cell responses elicited by different vaccine platforms are qualitatively distinct: the intracellular localization of RNA vaccines allows for MHC I presentation and generation of associated CD8^+^ T cell responses that is not typically observed in response to immunization with subunit vaccines (Figure 1). The utility of this is debated for COVID-19 in which a neutralizing antibody response is the most desired initial outcome, recent data indicates that generation of longer term T cell responses provides an extended benefit [20,21]. For chronic infections such as *Leishmaniasis* the generation of both CD4^+^ and CD8^+^ T cell responses may be the optimal profile for affording protection[22].

## 3. Current Challenges for Leishmania Vaccines

### 3.1. Development

Numerous diverse technological platforms have been explored as *Leishmania* vaccine candidates, including live-attenuated or whole-killed parasites (first generation), recombinant proteins (second generation) and DNA vaccines (third generation). Theavailability of genetic information has significantly aided the development process. Several live attenuated *Leishmania* species have been rendered by genetic modification of critical parasite virulence or survival genes, perhaps most notably the Centrin gene-deleted series that includes *L. braziliensis, L. donovani*, *L. major and L. mexicana* parasites [23]. Significant effort has also been focused on defining antigens associated with a protective immune response against *Leishmania*. Selected targets have been delivered as formulated protein, or as DNA sequences either alone or within vectors such as adenoviruses or even within *Leishmania* themselves, and measurements have included cellular immune responses and protection [23,24,25,26,27,28,29,30,31]. These studies clearly demonstrate that delivery of defined antigens (or antigenic sequences) in a manner that induces appropriate T cell responses affords protection in animals. Although single antigens may prove to be effective vaccines it is possible, especially when attempting to induce protection against multiple parasite species, that a multi-antigen approach would be desirable. Several defined subunit vaccines consisting of recombinant fusion proteins formulated with adjuvant that elicit protective Th1 responses have been developed [32,33,34]. Although this reduces manufacturing costs, further attempts should be made to address the practical aspects of vaccine production during early development. Case in point is the M72 tuberculosis vaccine candidate that was produced with a scientifically sound approach but which failed to advance beyond phase 2 clinical trials due to costs of production and limited availability of adjuvant components. 

RNA technology has the potential to provide an effective and practical solution to vaccine development for a multitude of diseases, including many neglected tropical diseases [14]. RNA vaccine development requires only the target gene sequence be known and removes the need for pathogen culture or scaled recombinant protein production. Due to activation of various pattern-recognition receptors, RNA vaccines can be very immunogenic and have demonstrated a capacity for rapid induction of antibody responses to several emerging pathogens [16,35,36,37,38,39]. From their initial conception, by mimicking immunization with a live vaccine, nucleic acid vaccines, delivered virally, such as with viral replicon particles or similar systems have also held promise as an effective way to induce T cell immunity. While mRNA vaccines are translated directly from the incoming RNA molecules, introduction of repRNA into cells initiates ongoing biosynthesis of antigen-encoding RNA that results in dramatically increased transcription and hence greater protein yields for each RNA molecule delivered [40,41,42]. In addition, repRNA vaccines mimic an alphavirus infection in that viral-sensing stress factors are triggered and innate pathways are activated through Toll-like receptors (TLR) and retinoic acid inducible gene (RIG)-I to produce interferons, pro-inflammatory factors and chemotaxis of APCs, as well as promoting antigen cross-priming. As a consequence, repRNA typically elicit stronger immune responses than similar quantities of mRNA, or equivalent responses when provided at substantially lower doses [42]. 

The first studies of mRNA and repRNA in the context of immunization demonstrated antigen-specific cell-mediated and humoral-adaptive immune responses against the influenza A [43,44,45]. Unlike for the rapid initiation of antibody responses, however, mRNA and defined subunit vaccines (antigen and adjuvant) typically require multiple administrations over an extended period of time to raise effective T cell responses. Viral delivery of replicon RNA derived from the alphavirus genus has demonstrated potent CD8^+^ T cell responses, as for example in mice immunized with a naked repRNA derived from a vaccine strain of the alphavirus Venezuelan equine encephalitis virus (VEEV), TC-83, which has a long history in pre-clinical and, more recently, clinical development [46,47,48]. 

In contrast to the relative ease in generating responses in mice (where even naked RNA can generate immunity if larger enough doses are provided), several formulation strategies have failed upon evaluation in primates and humans [49]. Thus, although many candidates may appear valid in small animal models this “primate barrier” represents a critical hurdle to clinical use of RNA vaccines. Accordingly, different vehicles have been developed for the RNA to both protect the molecule from degradation and improve cell transfection/uptake [50,51,52]. In vitro mRNA transfection of various eukaryotic cells using a lipid nanoparticle (LNP) was demonstrated in 1989 and in 2007, de Jong et al. demonstrated that LNP encapsulated antigen can induce a strong immune response and enhance immune efficacy [53,54]. LNP have continued to be a key consideration in RNA vaccine development [40,55]. Clinical trials have demonstrated the dose-dependent safety and tolerability of LNP-based nucleic acid vaccines is suitable for a pandemic response, although improved safety profiles would appear to be desirable for non-emergency situations [56]. Allergic reactions have been attributed to polyethylene glycol and polysorbate excipients within these vaccines, and post-licensure safety monitoring has observed increased risk of myo- and pericarditis in recipients that have prompted updates to regulatory guidelines for trial conductance [57,58,59,60,61]. As an alternate delivery formulation, the LION^TM^ family of highly stable cationic emulsions was developed. LION^TM^ enables electrostatic association with RNA molecules when combined by a simple 1:1 (v/v) mixing step and the formulation is colloidally stable for at least 12 months when stored at 4 and 25 °C ([62], and unpublished data). Unlike unformulated repRNA, when formulated with LION^TM^ repRNA molecules are protected from RNase-catalyzed degradation and ongoing clinical trials (clinicaltrials.gov NCT05132907, NCT04844268) are supporting the safety of LION^TM^ formulated repRNA vaccines (unpublished data).

Building upon a long-term antigen selection program that encompassed cutaneous and visceral leishmaniasis patients, New and Old World leishmaniasis, and associated *Leishmania* species, the sequences of the LEISH-F2 and LEISH-F3+ fusion proteins were selected for evaluation as repRNA vaccine candidates capable of raising protective T cell responses. In the direct comparison of subunit and repRNA vaccines with these same antigen-inserts, the T cell responses elicited were qualitatively distinct: repRNA elicited a CD8^+^ T cell response that was not observed in animals immunized with subunit vaccine. Further, priming with the repRNA followed by boosting with subunit vaccine generated extremely potent CD4^+^ T cell responses (at levels greater than those achieved with either modality on its own) and provided immunity sufficient to protect against *L. donovani* challenge [63]. These data indicate the important impact that the antigen production/ presentation platform can have on immunity and suggest that RNA immunization can be used for the preferential induction of T cell responses. It should also be noted that our previous mouse validation study used naked RNA replicon (i.e., formulated only in a saline diluent), with the knowledge that a previous restrictive feature of naked RNA vaccines was the transition to use with a clinically appropriate formulation. As discussed previously, advancements in formulation technology strongly suggest that both the relative immunogenicity and stability of the target-specific RNA can be substantially enhanced by appropriate formulation. Efficient introduction of genetic material may also lead to an extended period of antigen presentation/persistence relative to the exogenous delivery of protein antigen achieved with defined subunit vaccines, and this may facilitate the generation of memory T cell responses. Continued monitoring of participants in COVID-19 vaccine trials will be informative in determining the durability of the T cell responses that both mRNA and self-amplifying/ replicon RNA platforms induce. Long term monitoring is also required to determine fulminant safety profiles for each RNA platform, with contextualization against other authorized vaccines (adenoviruses and defined subunit) to identify if any concerns are associated with the platform or are due to the immunizing Spike antigen.

### 3.2. Clinical Evaluation

Addressing the on-the-ground reality of inconsistent pockets of local *Leishmania* transmission within much larger overall endemic regions, establishing controlled human infection models (CHIM) provides an opportunity to evaluate vaccines in the context of assured infection rates, with a marked impact in terms of reducing study complexity and cost. For example, estimates suggest that use of a CHIM for sand fly transmitted *Leishmania major* may require as few as 30 subjects per arm to detect vaccine efficacy of 60%. This compares favorably to the need for many hundreds if not thousands of subjects in conventional natural exposure trials. CHIM studies, by virtue of the known time of exposure, also provide an excellent opportunity to identify immunological correlates of protection and to further understand disease pathogenesis. Such findings from CHIM studies can then be evaluated and used to interpret data within larger field trials. CHIM studies and prospective human infection studies exist for a diverse array of pathogens, including for SARS-CoV-2 [64,65,66,67,68,69,70]. Importantly, CHIM can have surprising outcomes, such as a CHIM for malaria that did not fulfill the hypothesis that strong cellular immune responses impacted parasite growth rates but rather directed focus to achieving sufficient antibody titers [71].

It is important to note that experimental infection of humans with *Leishmania* spp. is extremely well established for both needle challenge and sand fly initiated infection [72,73]. Many factors weigh on the decision to develop a CHIM, however, including (a) poor disease control and impact on morbidity and mortality; (b) lack of successful vaccines, despite a number of candidate vaccines/antigens in the development pipeline; (c) absence of effective treatments and/or evidence of drug resistance; and perhaps most important, (d) a defined and treatable pathogen strain or species relevant to clinical disease. Leishmaniasis appears to satisfy each of these criteria: despite vector control efforts, highly endemic regions persist and cause suffering; several *Leishmania* species have been characterized genetically and multiple antigens have been proposed as vaccine candidates; despite improving drugs, drug resistance has and continues to emerge; and finally, establishing well characterized, and preferably GMP compliant, parasite banks for clinical use is being addressed [74]. Beyond establishing the rational and required tools, several practical and safety concerns with any proposed *Leishmania* CHIM then present themselves. A study involving non-infected sand fly biting was used to establish parameters for challenge and importantly to gauge and incorporate public perceptions of this type of study into a challenge protocol ([75,76] and clinicaltrials.gov: NCT03999970). A clinical study to evaluate the reproducibility of a CHIM for sand fly transmitted cutaneous leishmaniasis has similarly gained ethical and institutional approval and is ongoing (clinicaltrials.gov: NCT04512742). Despite the upfront costs of establishing the CHIM, their use may ultimately represent a cost-effective strategy for prioritizing vaccine candidates because one of their most significant benefits may be in preventing candidates that perform poorly from advancement into large scale clinical trials where they would otherwise consume both investigators and potential recruits in endemic regions. It is important to emphasize that CHIM may not completely replace traditional efficacy trials as their limitations include the participation of individuals in non-endemic regions such that typical environmental pressures are different (i.e., continued or multiple low level exposures that do not establish infection but could influence underlying immunity), ethnic/ genetic varianaces, and, given that *Leishmaniases* are neglected tropical diseases that disproportionately impact the poor, socioeconomic status and underlying health status. If these *Leishmania* CHIM efforts are successful, however, they could well become an important stepping-stone for leishmaniasis candidate vaccines before evaluation in larger field trials (Figure 2). In addition, data from CHIM studies has directly led to vaccine licensure or vaccine usage policy changes and a similar approach might be applicable for vaccines against forms of leishmaniasis where field trials would be almost impossible [77].

An additional alternative, or adjunct, to awaiting primary infection and disease development is to evaluate vaccine candidates among infected individuals that can be provided a vaccine post-infection. Speed to, and completeness of, clinical cure and parasite resolution can then be used as an indication of efficacy. If the clinical course is mild and there is a “window of opportunity” that is ethically acceptable (such as was the case in the evaluation of the ChAd63-KH vaccine as a therapeutic for persistent post-kala azar dermal leishmaniasis (PKDL)) vaccines may be tested as stand alone treatments[26]. In more severe presentations, however, this can be confounded by the ethical requirement for provision of standard care to affected individuals and it is therefore important to establish that the vaccine of interest is compatible with chemotherapy. In the case of visceral leishmaniasis, an additional clinical endpoint of sequelae such as post-kala azar dermal leishmaniasis could be used.

## 4. Leverage of a Profitable Veterinary Application?

In contrast to the human situation, veterinary vaccines based on *Leishmania* parasite lysates have been advanced to approval for canine leishmaniasis (CanL). Injection of a vaccine comprising total antigens of *Leishmania amazonensis* plus saponin (LaSap) to infected dogs alleviated clinical symptoms and reduced parasite loads in the skin for at least 6 months [78]. Post-infection use of recombinant *Leishmania* A2 protein plus saponin (LeishTec®) has also indicated a 25% reduction in risk of developing CanL in asymptomatic dogs and a 70% reduction in mortality among younger dogs (<6 years old) [79]. Similarly, immunotherapeutic investigation of a vaccine comprising *Leishmania braziliensis* antigens and TLR4 agonist MPL (LBMPL vaccine) demonstrated reductions in both parasite burden and the intensity of disease in the treated dogs, accompanied by a blocking of their transmission of *L. infantum* to sand flies (observed in 66% (6 of 9) dogs evaluated 3 months later [80]. 

It is well established that infection and clinical status of each *Leishmania*-affected dog influences the response to treatment and study outcomes could therefore be impacted by infection level at time of treatment. Supplementation of subunit vaccines with MPL has promoted success, or contrarily has had no impact, in halting disease progression [81,82,83]. Another consideration is onset of immune exhaustion that precedes the transition from asymptomatic *Leishmania* infection to progressively worsening CanL [84]. Ex vivo incubation with combinations of TLR4, 7, and 7/8 agonists allowed the identification of robust Th1 cells from symptomatic dogs and further suggested the possibility that *Leishmania*-induced cellular exhaustion could be overcome by potent immunization [85]. The involvement of TLR7 in this is particularly relevant for repRNA vaccines that can engage this receptor [63].

A further complication in providing effective immune therapy may arise from the observation that even after ‘cure’ parasites are retained in very low numbers and maintain immune status. The hypothesis that allopurinol-induced parasite reduction in CanL-affected dogs would render animals more able to develop robust immunity to provide additional longer term control of infection, was tested by evaluating *L. infantum* infected dogs for clinical and parasitological outcomes following short-term treatment with allopurinol, either alone or in combination with a defined subunit vaccine. Dogs treated with allopurinol alone alleviated their CanL symptoms but had only a transient reduction in parasites that rebounded after a few months. Concomitant immunization with Leish-F2 + SLA-SE, however, not only improved clinical status but also elicited *L. infantum* clearance from lymphoid tissues and systemic organs in the long term (Figure 3 and [86]). 

## 5. Conclusions

As we have outlined, RNA vaccines could address several current limitations in generating a truly field applicable vaccine for leishmaniasis. RNA vaccines can be made more cost-effectively than either defined subunit or vector-based vaccines and, given the growing familiarity and acceptance of RNA platforms with regulatory agencies, progression to trial is likely to become simpler and more rapid. In this way, we envision that trials can be planned and conducted in a manner more responsive to the identification of relatively small and transient “hot-spots”, permitting a much more efficient design for the evaluation of efficacy. As an alternative, proof-of-concept of safety and efficacy could be generated in human infection models or inferred among patients undergoing therapy.

## Figures and Tables

**Figure 1 pathogens-11-01259-f001:**
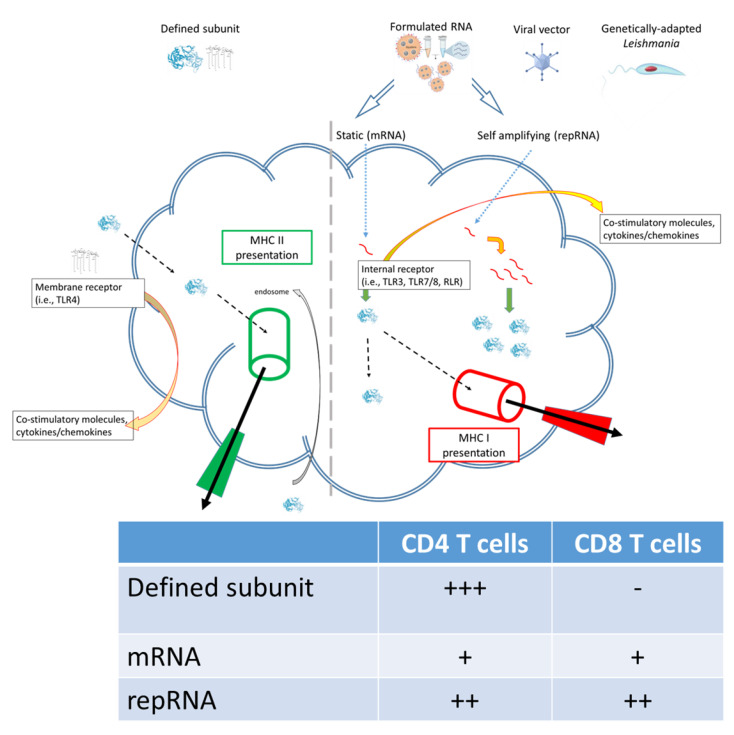
Subunit and RNA replicon vaccines can generate different quality of immune response.

**Figure 2 pathogens-11-01259-f002:**
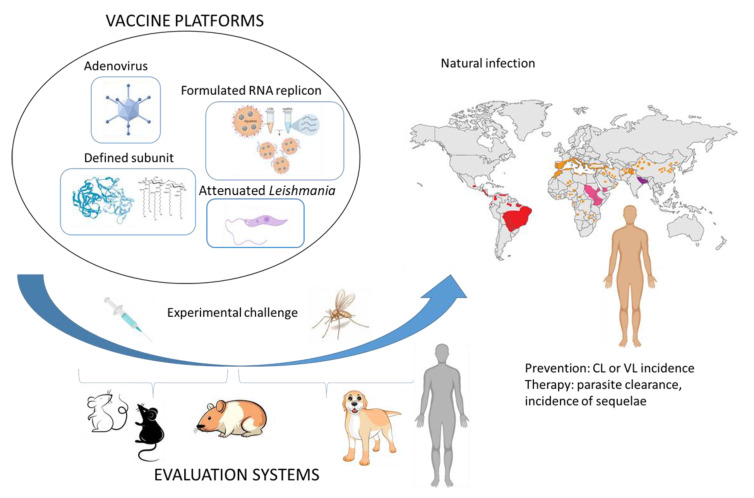
Vaccine classes and how they can be evaluated for efficacy against Leishmaniasis.

**Figure 3 pathogens-11-01259-f003:**
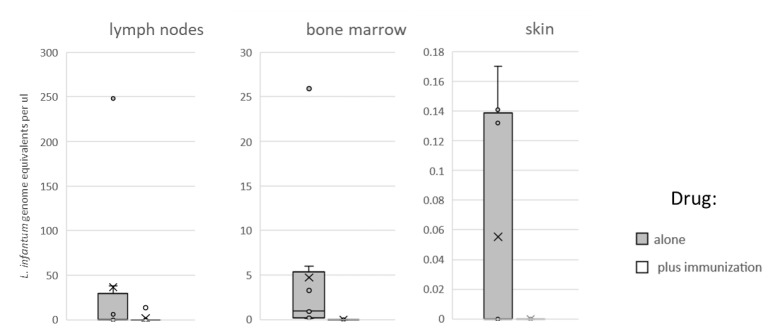
Addition of vaccine to drug treatment generates sustained *L. infantum* clearance. Infected dogs were treated with either drug (allopurinol) alone or drug plus immunization (Leish-F2 + SLA-SE) then one year later *L. infantum* burden determined in the indicated organs were determined. Adapted from original data published in [86].

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
