# Peer review of "Leishmaniasis Vaccines: Applications of RNA Technology and Targeted Clinical Trial Designs"

_pathogens, 2022, doi:10.3390/pathogens11111259_

Round 1

Reviewer 1 Report

The authors of the manuscript titled ‘ Leishmaniasis Vaccines : Applications of RNA Technology and Targeted Clinical Trial Designs ’ report on the way forward with vaccine development for Leishmaniasis. The article succinctly summarize the current treatment approaches and use the challenges therein to convincingly argue for RNA based vaccines. The article is focused and lucidly written, with appropriate references.

Author Response

No edits recommended. Many thanks.

Reviewer 2 Report

The authors discuss the possibility of the use of a new approach to vaccines against Leishmania parasites. They explain the difficulties to develop vaccines and the gaps that exist in Leishmania vaccines.  I think that the author developed very well the theme.

Author Response

No edits recommended. Many thanks.

Reviewer 3 Report

It is well written overall, and gives really useful information on RNA vaccines and CHIM to the readers. On the contrary, description on the potential of RNA vaccines for leishmaniasis is limited. In fact, there is only one paper (Reference 61) cited as a genuine example of RNA vaccines for leishmaniasis in this manuscript. If it is truly the only one in the literature, it is too early to say RNA technology is beneficial to address the bottlenecks encountered for Leishmania vaccines.

Figure 1 is not very clear. The figure title mentions only subunit and RNA vaccines, as is the table, but the illustrations contain viral vector and genetically adapted Leishmania as well. If you do not include them for comparison with subunit and RNA vaccines, it is better to omit them from the figure whereas they are known vaccination tools for leishmaniasis.

Line 77: One might say that protein production can also be done in a cell-free in vitro system (e.g., wheat germ expression system), so slight modification to this sentence is preferred.

L87: It is too much to say that CD8+ T cells are not elicited by subunit vaccines (e.g., adjuvant/delivery system). Again, minor modification is preferred.

L209: experimental infection with Leishmania not leishmaniasis

Author Response

Figure 1 is not very clear. The figure title mentions only subunit and RNA vaccines, as is the table, but the illustrations contain viral vector and genetically adapted Leishmania as well. If you do not include them for comparison with subunit and RNA vaccines, it is better to omit them from the figure whereas they are known vaccination tools for leishmaniasis. - depiction of viral vector and genetically adapted Leishmania removed as suggested.

Line 77: One might say that protein production can also be done in a cell-free in vitro system (e.g., wheat germ expression system), so slight modification to this sentence is preferred. - Updated to read "One considerable logistical advantage of RNA-based vaccines over the majority of other platforms is that the RNA can be produced in a cell-free environment by in vitro transcription.."

L87: It is too much to say that CD8+ T cells are not elicited by subunit vaccines (e.g., adjuvant/delivery system). Again, minor modification is preferred. - updated to read "...generation of associated CD8+ T cell responses that is not typically observed in response to immunization with subunit vaccines" 

L209: experimental infection with Leishmania not leishmaniasis - corrected to Leishmania spp.

Reviewer 4 Report

This review discusses the potential use of RNA technology for a development of a vaccine for leishmaniasis and using controlled human infection model (CHIM) for testing it. RNA technology has been highly effective for developing COVID19 vaccine and currently been explored for other viral vaccines and repRNA could be an appropriate platform.  CHIM models are also being used for testing vaccines for parasitic diseases such as malaria. Overall, this is a well written review which is timely and thought provoking and is acceptable. However, some discussion on potential drawbacks if any would benefit.

Author Response

Reviewer 4:

This review discusses the potential use of RNA technology for a development of a vaccine for leishmaniasis and using controlled human infection model (CHIM) for testing it. RNA technology has been highly effective for developing COVID19 vaccine and currently been explored for other viral vaccines and repRNA could be an appropriate platform.  CHIM models are also being used for testing vaccines for parasitic diseases such as malaria. Overall, this is a well written review which is timely and thought provoking and is acceptable. However, some discussion on potential drawbacks if any would benefit.

Response:

We thank the reviewer for these comments. As requested, we have added some discussion on potential drawbacks.

Reviewer 5 Report

Leishmaniasis is a neglected tropical disease impacting large number of people in the tropical and subtropical world causing significant morbidity and mortality. Currently there is no licensed human vaccine against this disease. There are a few candidates, using different approaches, at various stages of preclinical and clinical trials. This  review article describes another approach i.e., use of RNA technology in Leishmania vaccine development recently pioneered for SARS COV -19 virus vaccine. In addition, this paper also describes  strategies  for the clinical trials such as Controlled Human Infection Model (CHIM) to evaluate Leishmania vaccines. The proposed RNA technology has been previously tested in the authors laboratory but needed heterologous boosting with recombinant Leishmania proteins and   adjuvants to have better adaptative immune response.

Overall, the review has merit, but the authors need to highlight the limitations of the strategies which they propose for both vaccine development as CHIM. For example, unlike SARS-COV19 vaccine which is solely dependent on generating humoral response based  one or two viral proteins, Leishmania vaccine will need more that a few antigens put together to generate a potent long lasting cellular response that has been shown in previous studies from healed leishmaniasis people. Similarly, there are limitations in applicability of CHIM in species of Leishmania which cause visceral or mucocutaneous diseases.

Following are the specific comments:

1.       Lines 78-80 This is not entirely accurate, recent studies [PMID: 36138186, PMID: 35981045] demonstrated that while nAbs are important, long-term protection seems to require CD8 T cell memory response, even in instances of diminishing antibody titers. Need to revise the text to reflect these clinical data. 

2.       While the advent of mRNA based vaccine technology simplified the challenges of scale and deployment as noted by the authors, the technology is yet to be tested whether durable CD8/CD4 T memory responses can be induced. So far studies in COVID vaccines show that mRNA vaccines show efficacy for 6 months to 1 year requiring booster doses thereafter. Thus, while mRNA vaccine remains a fascinating platform, the durable protective immunity is by no means foregone conclusion. These challenges need to be highlighted.

3.       RNA replicon elicited CD8 T cell responses in a heterologous regimen in a L donovani challenge in a study referenced by the authors previous publication (Front. Immunol. 2018) in support of mRNA vaccines. However, it is far from certain if these responses are sufficient for durable protection and need to be demonstrated in adaptive transfer experiments.

4.       In the study referenced in #3 above , (Fig.6) protection against L. donovani infection in mice using RNA + AG+ Adjuvant as vaccine did not show any significant parasite control.  Therefore, it is not clear how combination of both RNA based vaccines boosted with a few proteins as proposed in the review  is going to be efficacious. 

5.       The authors also need to describe other strategies of Leishmania vaccine development  so that the reader has better overall perspective of the current status of the Leishmania vaccine development. 

6.       Safety concerns: the authors need to also highlight the safety concern with mRNA technology based vaccines as has been observed in COVID vaccines.

7.       Both the topics described in the review are of high importance. However, to a reader it looks like that two separate thoughts (mRNA vaccine and strategies for clinical trials) have been lumped together. It would do justice to the field of Leishmania vaccines if they were two separate reviews describing in detail both the pros and cons of two topics.

Author Response

Reviewer 5:

Leishmaniasis is a neglected tropical disease impacting large number of people in the tropical and subtropical world causing significant morbidity and mortality. Currently there is no licensed human vaccine against this disease. There are a few candidates, using different approaches, at various stages of preclinical and clinical trials. This  review article describes another approach i.e., use of RNA technology in Leishmania vaccine development recently pioneered for SARS COV -19 virus vaccine. In addition, this paper also describes  strategies  for the clinical trials such as Controlled Human Infection Model (CHIM) to evaluate Leishmania vaccines. The proposed RNA technology has been previously tested in the authors laboratory but needed heterologous boosting with recombinant Leishmania proteins and   adjuvants to have better adaptative immune response.

Overall, the review has merit, but the authors need to highlight the limitations of the strategies which they propose for both vaccine development as CHIM. For example, unlike SARS-COV19 vaccine which is solely dependent on generating humoral response based  one or two viral proteins, Leishmania vaccine will need more that a few antigens put together to generate a potent long lasting cellular response that has been shown in previous studies from healed leishmaniasis people. Similarly, there are limitations in applicability of CHIM in species of Leishmania which cause visceral or mucocutaneous diseases.

Response:

We thank the reviewer for these comments, and have inserted point-by-point responses below.

Following are the specific comments:

  1. Lines 78-80 This is not entirely accurate, recent studies [PMID: 36138186, PMID: 35981045] demonstrated that while nAbs are important, long-term protection seems to require CD8 T cell memory response, even in instances of diminishing antibody titers. Need to revise the text to reflect these clinical data. 

Response:

We thank the reviewer for identifying these recent publications and their relevance to this review. The sentence(s) have been updated and references cited.

  1. While the advent of mRNA based vaccine technology simplified the challenges of scale and deployment as noted by the authors, the technology is yet to be tested whether durable CD8/CD4 T memory responses can be induced. So far studies in COVID vaccines show that mRNA vaccines show efficacy for 6 months to 1 year requiring booster doses thereafter. Thus, while mRNA vaccine remains a fascinating platform, the durable protective immunity is by no means foregone conclusion. These challenges need to be highlighted.

Response:

As suggested we have added explicit comment on durability which will require continued clinical monitoring of RNA vaccine recipients to be fully elucidated. This comment is somewhat addressed by the reviewers first comment.

  1. RNA replicon elicited CD8 T cell responses in a heterologous regimen in a L donovani challenge in a study referenced by the authors previous publication (Front. Immunol. 2018) in support of mRNA vaccines. However, it is far from certain if these responses are sufficient for durable protection and need to be demonstrated in adaptive transfer experiments.

Response:

The reviewer’s emphasis on durability of responses is appreciated and, where applicable, we have indicated the need for this to be determined.

  1. In the study referenced in #3 above , (Fig.6) protection against L. donovani infection in mice using RNA + AG+ Adjuvant as vaccine did not show any significant parasite control.  Therefore, it is not clear how combination of both RNA based vaccines boosted with a few proteins as proposed in the review  is going to be efficacious. 

Response:

Our published data indicated that a repRNA prime followed by an Ag/adjuvant boost significantly reduced L. donovani burden in mice (p-value < 0.05; 6 of 7 immunized mice having parasite burdens lower than the mean of unimmunized controls)). Further, we can clear in our discussion that these data a. “indicate the important impact that the antigen production/ presentation platform can have on immunity and suggest that RNA immunization can be used for the preferential induction of T cell responses” and b. have the limitation that the repRNA was not formulated when that data were generated.

We do not discuss using combination RNA-based vaccines with protein boosts.

  1. The authors also need to describe other strategies of Leishmania vaccine development  so that the reader has better overall perspective of the current status of the Leishmania vaccine development. 

Response:

Alternate strategies are discussed in section “Current challenges for Leishmainia vaccines, Development”

  1. Safety concerns: the authors need to also highlight the safety concern with mRNA technology based vaccines as has been observed in COVID vaccines.

Response:

Comment has been added with regard to the need for continued monitoring of mRNA and sa/repRNA recipients.

  1. Both the topics described in the review are of high importance. However, to a reader it looks like that two separate thoughts (mRNA vaccine and strategies for clinical trials) have been lumped together. It would do justice to the field of Leishmania vaccines if they were two separate reviews describing in detail both the pros and cons of two topics.

Response:

We appreciate the reviewer’s identification of these two key areas of discussion. We believe that vaccine manuscripts/reviews are typically focused upon the preclinical development without consideration of clinical development strategies, and thus determined to blend the two elements herein to provide a consolidated and cohesive submission.